# Serious adverse events reported with benzimidazole derivatives: A disproportionality analysis from the World Health Organization's pharmacovigilance database

**Pamella Modingam[1], Jean-Luc Faillie[2,3], Jérémy T. Campillo[1] ***

**1** TransVIHMI, Université de Montpellier, INSERM Unité 1175, Institut de Recherche pour le Développement (IRD), Montpellier, France, **2** CHU Montpellier, Department of Medical Pharmacology and Toxicology, Montpellier, France, **3** Desbrest Institute of Epidemiology and Public Health, UMR UA11 University of Montpellier – INSERM, Montpellier, France

* jeremy.campillo@ird.fr

**Data Availability Statement:** Data cannot be shared publicly because of restriction of access to

## Abstract

### Background

Benzimidazole derivatives are widely used anthelmintic drugs, particularly in mass campaigns for intestinal parasitosis treatment. Despite their generally good safety profile, serious adverse reactions have been reported. This study aims to identify potential pharmacovigilance signals for benzimidazole derivatives using disproportionality analysis in the WHO database.

### Methodology

A case/non-case study was conducted using data from the WHO VigiBase database (2000–2023). Cases were individual case safety reports (ICSRs) where at least one suspected serious adverse event of interest was reported, while non-cases were ICSRs reporting any adverse events other than the serious adverse events of interest. Reporting Odds Ratios (RORs) and 95% confidence intervals were calculated to assess disproportionate reporting. Analyses were adjusted for potential confounding factors and a sensitivity analysis with imputed missing data was performed.

### Principal findings

Among 19,068 serious reports analyzed, significant disproportionality signals were found for benzimidazole derivatives compared to other anthelmintic drugs, notably for bone marrow failure and hypoplastic anemia (adjusted ROR 9.44 [5.01–18.9]), serious leukopenia (3.89 [2.64–5.76]), serious hepatic disorders (3.10 [2.59–3.71]), hepatitis (2.88, 95% CI 2.29–3.63) and serious urticaria (2.02, 95% CI 1.36–2.99). We have also highlighted a new signal not mentioned in the summaries of product characteristics for seizures with benzimidazole

VigiBase. Data are available from the WHO Global Individual Case Safety Report (ICSR) database, VigiBase for researchers who meet the criteria for access to confidential data. Vigibase has the data we used and makes them available to people working in a hospital pharmacovigilance service via the email address CustomSearches@who-umc.org.

**Funding:** The author(s) received no specific funding for this work.

**Competing interests:** The authors have declared that no competing interests exist.

derivatives. Secondary analysis revealed these signals were primarily reported with albendazole.

## Conclusions/Significance

This study identified potential pharmacovigilance signals for serious hematological and hepatic adverse events for benzimidazole derivatives, particularly albendazole. New signal for benzimidazole derivatives has been described for seizures. These findings underscore the need for vigilant monitoring during benzimidazole derivatives use and warrant further pharmacoepidemiologic studies to confirm these signals and investigate underlying mechanisms.

## Author summary

Benzimidazole derivatives, including albendazole and mebendazole, are widely used anthelmintics known for their broad spectrum of activity and general tolerability. Some adverse reactions have been reported with benzimidazole derivatives, including hematological, immune, neurological, gastrointestinal, hepatobiliary, and skin disorders. This is the first analysis of serious adverse events reported with benzimidazole derivatives using the WHO global pharmacovigilance database, VigiBase. The study identifies potential pharmacovigilance signals for benzimidazole derivatives, including increased reporting of bone marrow failure, hypoplastic anemia, leukopenia, and hepatic disorders. It also reveals that albendazole is more frequently reported in several serious adverse events compared to other benzimidazole derivatives and anthelmintic drugs, particularly for hematological and hepatotoxic effects. We have also highlighted a new signal, not mentioned in the summaries of product characteristics, for benzimidazole derivatives. These findings highlight the importance of careful monitoring when using benzimidazole derivatives and suggest the need for further studies to confirm these observations and explore the underlying mechanisms.

## Introduction

Anthelmintic drugs primarily include benzimidazole derivatives, ivermectin (IVM), and praziquantel (PZQ). These drugs are known for their broad spectrum of anthelmintic activity, effectiveness, and ease of use. The main benzimidazole derivatives used in human medicine include albendazole (ALB), flubendazole (FLU), triclabendazole (TRI), mebendazole (MEB), and thiabendazole (THIA) [1]. ALB and MEB are used in annual deworming campaigns for soil-transmitted helminth infections (roundworm, whipworm, and hookworm) in at-risk populations (preschool children, school-age children, women of reproductive age, and adults in certain high-risk occupations) living in endemic areas [2,3]. It is estimated that several hundred million doses of ALB and MEB have been administered since the inception of these control campaigns.

The broad spectrum of activity of benzimidazole derivatives is attributed to their ability to inhibit tubulin polymerization. This mechanism disrupts the biological processes of digestive tract parasites, including adults and larvae, leading to biochemical changes that result in parasite immobilization and death. In addition to its use in mass drug administration campaigns,

ALB is indicated for a wide range of parasitic infections, while MEB and FLU are used for intestinal nematodes, and TRI for fascioliasis. THIA, formerly used for tissue nematodes and strongyloidiasis, was withdrawn from the French market due to high toxicity. PZQ is used for trematode and cestode infections, while IVM is indicated for various nematode infections and scabies [4–8].

While benzimidazole derivatives are generally considered well-tolerated, some adverse reactions have been reported, including hematological, immune, neurological, gastrointestinal, hepatobiliary, and skin disorders. These are documented in the official information and summaries of product characteristics (SPCs) in most countries.

Safety profiles of some anthelmintic drugs have already been analyzed using disproportionality analysis of the WHO pharmacovigilance database, including PZQ [9], levamisole [10] and IVM [10–12]. However, despite their widespread use globally, to our knowledge, no disproportionality study has yet been conducted to assess the overall safety profile of benzimidazole derivatives and potentially detect new adverse events reported with this class.

Following a review of adverse reactions reports related to benzimidazole derivatives published in the literature, we investigated the WHO global pharmacovigilance database, VigiBase, for all suspected serious adverse events reported after anthelmintic treatment. We then conducted disproportionality analyses comparing benzimidazole derivatives to PZQ and IVM. Specifically, the aims of this study were: (i) to identify new potential pharmacovigilance signals (increased reporting of suspected adverse events after treatment with benzimidazole derivatives compared to PZQ and IVM), and (ii) to compare the safety profile of ALB to other benzimidazole derivatives.

## Methods

### Data source

Deduplicated data were specifically provided by the Uppsala Monitoring Centre upon request from the World Health Organization (WHO) Global Individual Case Safety Report (ICSR) database, VigiBase [13]. This database contains over 38 million cases of suspected adverse events reported by national pharmacovigilance centers from more than 130 countries participating in the WHO Program for International Drug Monitoring [14]. An ICSR is an anonymized report for a single individual who experienced adverse event(s) potentially linked to the use of one or more drugs. Each ICSR contains sociodemographic information (age, sex, reporter qualification, country of origin, year of report), drug administration details (frequency, dosage, co-medications), and information about the reported adverse event(s). The latter includes the seriousness criteria as defined by the International Council for Harmonization of Technical Requirements for Pharmaceuticals for Human Use (ICH) [15], verbatim description of the adverse event, and associated terms from the Medical Dictionary for Regulatory Activities (MedDRA) developed by the ICH [16].

### Study design

We conducted disproportionality analyses using the case/non-case method to identify disproportionate reporting, defined as a higher-than-expected number of adverse events reports compared to other events recorded in the database. This was achieved by calculating Reporting Odds Ratios (RORs), which compare the odds of exposure to benzimidazole derivatives between cases and non-cases [17,18]. All ICSRs of serious adverse events reported with anthelmintic drugs between January 1, 2000, and December 31, 2023, were extracted from VigiBase. An ICSR can contain multiple adverse events. Duplicates and reports involving the concomitant administration of multiple anthelmintic drugs were excluded. Only ICSR in which

anthelmintic drugs were suspected or implicated in interactions were included. Exposure to benzimidazole derivatives was determined by the mention of ALB, MEB, FLU, THIA, or TRI in the ICSR. Exposure to other anthelmintic drugs was determined by the mention of IVM or PZQ.

## Drug exposure

Drugs were identified primarily by their active ingredients to ensure consistency and comparability between different formulations and trade names. The selected anthelmintics were benzimidazole derivatives (MEB, THIA, ALB, FLU, TRI) (Anatomical Therapeutic Chemical codes P02CA, D01AC, and P02BX) and IVM, and PZQ (P02BA). The latter two were chosen as comparators due to their similar indications.

## Case definition

For each analysis, cases were ICSRs where at least one suspected serious adverse event of interest was reported, while non-cases were ICSRs reporting any adverse events other than the serious adverse events of interest. Each suspected serious adverse event of interest was identified by a selection of MedDRA terms (System Organ Class [SOC], High Level Term [HLT], and Preferred Term [PT]) reformulated into more specific terms (S1 Table). Serious adverse events of interest included hepatic disorders, hepatitis, cholecystitis, cholelithiasis, nausea and vomiting, cutaneous reactions, urticaria, toxidermias, drug reaction with eosinophilia and systemic symptoms (DRESS), Stevens-Johnson syndrome, toxic epidermal necrolysis, bullous conditions, acute generalized exanthematous pustulosis, allergic disorders, seizures, leucopenia, marrow depression and hypoplastic anemias, headaches, dizziness.

## Statistical methods

Only data concerning cases considered serious were used for statistical analysis. Descriptive statistics were used to summarize the basic characteristics of the ICSRs reporting serious adverse events after anthelmintic exposure according to sex, age group, geographic zone, reporting period, and reporter type, along with their numbers and frequency.

Our primary analysis consisted of calculating the ROR of each suspected adverse event of interest (and corresponding 95% confidence interval [95% CI]) for benzimidazole derivates compared to PZQ and IVM reported in VigiBase using logistic regression models. In the secondary analysis, adverse events were compared between ALB and all other benzimidazole drugs. Our analyses were adjusted for the potential confounders available in the line-listing extracted from Vigibase: sex, profession of the reporter (healthcare professional or not), age group (defined according to key life stages: 0–11 years, 12–17 years, 18–44 years, 45–65 years, and over 65 years), geographic area (Europe, Africa, Asia, North America, South America, Oceania), and reporting period (before 2016, 2016–2021, after 2021). To account for the presence of missing data, we performed a sensitivity analysis with imputation using the nearest neighbor's method (KNN method) [19]. Analyses were conducted using R version 4.3.3 software.

## Results

### Descriptive analysis of the serious ADRs reported with benzimidazole derivatives

Among the 257,898 ICSRs reporting adverse events after anthelmintic exposure registered in the database, 81,480 ICSRs that matched our criteria were reported from January 1, 2000, to

**Table 1. Characteristics of ICSRs containing serious adverse events reported with benzimidazole derivatives and others anthelmintics in VigiBase.**

| Characteristics (n, %) | Benzimidazole derivatives (n = 5,137) | Others anthelmintic drugs (n = 13,931) | Total (n = 19,068) |
|---|---|---|---|
| **Sex** | | | |
| Female | 2,463 (51.0%) | 6,897 (49.0%) | 4 889 (49.0%) |
| Male | 2,372 (49.0%) | 6,499 (51.0%) | 4 910 (51.0%) |
| Unknown | 302 | 535 | 837 |
| **Age group (years old)** | | | |
| 0–11 | 954 (21.0%) | 581 (5.1%) | 1,535 (9.6%) |
| 12–17 | 308 (6.9%) | 552 (4.8%) | 860 (5.4%) |
| 18–44 | 1,701 (38.0%) | 4,417 (38.0%) | 6,118 (38.0%) |
| 45–65 | 1,019 (23.0%) | 3,739 (33.0%) | 4,758 (30.0%) |
| >65 | 459 (10.0%) | 2,208 (19.0%) | 2,667 (17.0%) |
| Unknown | 696 | 2,434 | 3,130 |
| **Geographic area** | | | |
| Europe | 1,964 (38.0%) | 3,553 (26.0%) | 5,517 (29.0%) |
| Africa | 843 (16.0%) | 3,096 (22.0%) | 3,939 (21.0%) |
| Asia | 789 (15.0%) | 1,181 (8.5%) | 1,970 (10.0%) |
| North America | 1,451 (28.0%) | 5,439 (39.0%) | 6,890 (36.0%) |
| Oceania | 21 (0.4%) | 102 (0.7%) | 123 (0.6%) |
| South America | 69 (1.3%) | 560 (4.0%) | 629 (3.3%) |
| **Reporting period** | | | |
| Before 2016 | 2,041 (40.0%) | 4,769 (34.0%) | 6,810 (36.0%) |
| 2016–2021 | 2,277 (44.0%) | 5,741 (41.0%) | 8,018 (42.0%) |
| After 2021 | 819 (16.0%) | 3,421 (25.0%) | 4,240 (22%) |
| **Active ingredients name** | | | |
| Mebendazole | 1,425 (28.0%) | | 1,425 (7.5%) |
| Albendazole | 3,421 (67.0%) | | 3,421 (18.0%) |
| Flubendazole | 81 (1.6%) | | 81 (0.4%) |
| Tiabendazole | 156 (3.0%) | | 156 (0.8%) |
| Triclabendazole | 54 (1.1%) | | 54 (0.3%) |
| Ivermectin | | 13,011 (93.0%) | 13,011 (68.0%) |
| Praziquantel | | 920 (6.6%) | 920 (4.8%) |
| **Reporter type** | | | |
| No health professional | 1,106 (24.0%) | 3,260 (25.0%) | 4,366 (25.0%) |
| Health Professional | 3,478 (76.0%) | 9,757 (75.0%) | 13,235 (75%) |
| Unknown | 553 | 914 | 1,467 |

n, number of ICSRs

December 31, 2023. We excluded 49 duplicates and 391 cases with no available MedDRA terms. Of the remaining ICSRs, 19,068 contained a serious adverse event, of which 5,137 were reported after a benzimidazole derivative exposure (26.91%) and 13,931 cases were reported after another anthelmintic drug exposure (73.09%). Table 1 shows the characteristics of the ICSR reporting serious adverse events according to anthelmintic drug exposure by age group, sex, reporter type, geographical area, active ingredient name, and reporting period. In benzimidazole derivatives-exposed patients, the sex distribution was balanced, with 51% of cases being female. Adults aged 18–44 were the most represented (38%), followed by the 45–65 age group (23%) and children aged 0–11 (21%). Reports from Europe dominated with 38% of cases, followed by North America (28%). Africa and Asia represented 16% and 15% of cases,

**Table 2. Serious adverse events of interest according to anthelminthic drug exposure.**

| Serious adverse events (n, %) | Benzimidazole derivatives | Others anthelminthic drugs | Total |
|---|---|---|---|
| Hepatobiliary | | | |
| Hepatic disorders | 381 (7.42%) | 327 (2.35%) | 708(3.71%) |
| Hepatitis | 216 (4.20%) | 188 (1.35%) | 404(2.12%) |
| Cholecystitis and cholelithiasis | 8 (0.16%) | 6 (0.04%) | 14(0.07%) |
| Gastro-intestinal | | | |
| Serious nausea and vomiting | 306 (5.96%) | 405 (2.91%) | 711(3.73%) |
| Allergic and dermatological | | | |
| Cutaneous reactions | 586 (11.41%) | 1,430 (10.26%) | 2,016(10.57%) |
| Urticaria | 70 (1.36%) | 72 (0.52%) | 142(0.74%) |
| Toxidermias | 52 (1.01%) | 252 (1.81%) | 304(1.59%) |
| Drug reaction with eosinophilia and systemic symptoms | 3 (0.06%) | 39 (0.28%) | 42(0.22%) |
| Stevens-Johnson syndrome | 23 (0.45%) | 59 (0.42%) | 82(0.43%) |
| Toxic epidermal necrolysis | 8 (0.16%) | 62 (0.44%) | 70(0.37%) |
| Bullous conditions | 49 (0.95%) | 188 (1.35%) | 237(1.24%) |
| Acute generalized exanthematous pustulosis | 0 (0.00%) | 25 (0.18%) | 25(0.13%) |
| Serious allergic disorders | 71 (1.38%) | 217 (1.56%) | 288(1.51%) |
| Neurological | | | |
| Seizures | 73 (1.42%) | 79 (0.57%) | 152(0.80%) |
| Headaches | 154 (3.00%) | 387 (2.78%) | 541(2.84%) |
| Dizziness | 123 (2.39%) | 224 (1.61%) | 347(1.82%) |
| Hematological | | | |
| Leucopenia | 81 (1.58%) | 56 (0.40%) | 137(0.72%) |
| Marrow depression and hypoplastic anemias | 50 (0.97%) | 16 (0.11%) | 66(0.35%) |

n, number of ICSRs containing the adverse event of interest

A same ICSR can contain multiple adverse events of interest

respectively. Notifications before 2016 and for the period 2016–2021 represented 40% and 44% of cases, respectively. ALB exposure accounted for two-thirds of the serious cases (67%), followed by mebendazole exposure at 28%. Others benzimidazole derivatives (FLU, THIA, TRI) were marginal. Healthcare professionals reported 76% of the cases.

An analysis of literature data (S1 Material) and an initial analysis of the frequencies of adverse events reported with anthelmintics (S2 Table) enabled us to construct the list of effects of interest presented with their corresponding MedDRA terms in S1 Table. Table 2 describes the serious adverse events of interest according to anthelmintic drugs use. Serious cases reported after benzimidazole derivatives exposure reveals a preponderance of cutaneous reactions (11.41%), followed by hepatic disorders (7.42%) and nausea and vomiting (5.96%). No cases of acute generalized exanthematous pustulosis were reported after benzimidazole derivatives exposure.

## Disproportionality analyses

Table 3 shows the main disproportionality analyses of the serious adverse events of interest reported with benzimidazole derivatives compared to other anthelmintic drugs. The table also shows the results of the secondary analyses for each event of interest.

In the main analyses, a significant pharmacovigilance signal, in the sense of an over-reporting, was found with benzimidazole derivatives for the following serious adverse events, ranked

**Table 3. Disproportionality analysis of serious adverse events of interest reported after benzimidazole derivatives exposure.**

| Serious adverse events of interest | Cases* | Non-cases | Analysis 1[a] | | Analysis 2[b] | | Analysis 3[c] | |
|---|---|---|---|---|---|---|---|---|
| | | | ROR (IC95%) | P-value | aROR (IC95%) | P-value | aROR (IC95%) | P-value |
| **Hepatic disorders** | | | | | | | | |
| Benzimidazole derivatives | 381 | 4,756 | 3.33 (2.87–3.88) | <0.001 | 3.10 (2.59–3.71) | <0.001 | 3.23 (2.74–3.80) | <0.001 |
| Other anthelminthic drugs | 327 | 13,604 | Ref. | | Ref. | | Ref. | |
| Albendazole | 336 | 3,085 | 4.04 (2.98–5.62) | <0.001 | 5.70 (3.88–8.59) | <0.001 | 4.53 (3.22–6.49) | <0.001 |
| Other benzimidazole derivatives | 45 | 1,671 | Ref. | | Ref. | | Ref. | |
| **Hepatitis** | | | | | | | | |
| Benzimidazole derivatives | 216 | 4,921 | 3.21 (2.63–3.91) | <0.001 | 2.88 (2.29–3.63) | <0.001 | 2.95 (2.38–3.64) | <0.001 |
| Other anthelminthic drugs | 188 | 13,743 | Ref. | | Ref. | | Ref. | |
| Albendazole | 190 | 3,231 | 3.82 (2.58–5.92) | <0.001 | 4.82 (2.98–8.16) | <0.001 | 4.16 (2.70–6.66) | <0.001 |
| Other benzimidazole derivatives | 26 | 1,690 | Ref. | | Ref. | | Ref. | |
| **Cholecystitis et cholelithiasis** | | | | | | | | |
| Benzimidazole derivatives | 8 | 5,129 | 3.62 (1.26–11.0) | 0.017 | 1.89 (0.14–26.0) | 0.600 | 1.33 (0.30–6.29) | >0.900 |
| Other anthelminthic drugs | 6 | 13,925 | Ref. | | Ref. | | Ref. | |
| Albendazole | 1 | 3,420 | 0.07 (0.00–0.40) | 0.014 | 4,857,903(0.00-Inf) | >0.900 | 2.15 (0.04–192) | 0.700 |
| Other benzimidazole derivatives | 7 | 1,709 | Ref. | | Ref. | | Ref. | |
| **Nausea and vomiting** | | | | | | | | |
| Benzimidazole derivatives | 306 | 4,831 | 2.12 (1.82–2.46) | <0.001 | 1.50 (1.22–1.83) | <0.001 | 1.56 (1.31–1.85) | <0.001 |
| Other anthelminthic drugs | 405 | 13,526 | Ref. | | Ref. | | Ref. | |
| Albendazole | 197 | 3224 | 0.90 (0.71–1.15) | 0.400 | 0.61 (0.43–0.87) | 0.006 | 0.75 (0.56–1.01) | 0.060 |
| Other benzimidazole derivatives | 109 | 1,607 | Ref. | | Ref. | | Ref. | |
| **Cutaneous disorders** | | | | | | | | |
| Benzimidazole derivatives | 586 | 4,551 | 1.13 (1.02–1.25) | 0.023 | 0.81 (0.71–0.92) | 0.001 | 0.84 (0.75–0.94) | 0.002 |
| Other anthelminthic drugs | 1430 | 12,501 | Ref. | | Ref. | | Ref. | |
| Albendazole | 365 | 3,056 | 0.81 (0.68–0.97) | 0.019 | 0.64 (0.50–0.82) | <0.001 | 0.73 (0.59–0.91) | 0.006 |
| Other benzimidazole derivatives | 221 | 1495 | Ref. | | Ref. | | Ref. | |
| **Urticaria** | | | | | | | | |
| Benzimidazole derivatives | 70 | 5,067 | 2.66 (1.91–3.70) | <0.001 | 2.02 (1.36–2.99) | <0.001 | 1.92 (1.34–2.74) | <0.001 |
| Other anthelminthic drugs | 72 | 13,859 | Ref. | | Ref. | | Ref. | |
| Albendazole | 38 | 3,383 | 0.59 (0.37–0.95) | 0.030 | 0.56 (0.30–1.07) | 0.076 | 0.62 (0.35–1.10) | 0.100 |
| Other benzimidazole derivatives | 32 | 1,684 | Ref. | | Ref. | | Ref. | |
| **Toxidermias** | | | | | | | | |
| Benzimidazole derivatives | 52 | 5,085 | 0.56 (0.41–0.74) | <0.001 | 0.36 (0.25–0.51) | <0.001 | 0.33 (0.24–0.46) | <0.001 |
| Other anthelminthic drugs | 252 | 13,679 | Ref. | | Ref. | | Ref. | |
| Albendazole | 26 | 3,395 | 0.50 (0.29–0.86) | 0.012 | 0.23 (0.11–0.46) | <0.001 | 0.26 (0.13–0.51) | <0.001 |
| Other benzimidazole derivatives | 26 | 1,690 | Ref. | | Ref. | | Ref. | |
| **Drug reaction with eosinophilia and systemic symptoms** | | | | | | | | |
| Benzimidazole derivatives | 3 | 5,134 | 0.21 (0.05–0.57) | 0.009 | 0.14 (0.03–0.43) | 0.002 | 0.15 (0.03–0.42) | 0.002 |
| Other anthelminthic drugs | 39 | 5,134 | Ref. | | Ref. | | Ref. | |
| Albendazole | 1 | 3,420 | 0.25 (0.01–2.62) | 0.300 | 0.08 (0.00–1.69) | 0.130 | 0.08 (0.00–1.74) | 0.130 |
| Other benzimidazole derivatives | 2 | 1,714 | Ref. | | Ref. | | Ref. | |
| **Stevens-Johnson syndrome** | | | | | | | | |
| Benzimidazole derivatives | 23 | 5,114 | 1.06 (0.64–1.59) | 0.800 | 0.90 (0.51–1.57) | 0.700 | 0.79 (0.45–1.33) | 0.400 |
| Other anthelminthic drugs | 59 | 13,872 | Ref. | | Ref. | | Ref. | |
| Albendazole | 15 | 3,406 | 0.93 (0.41–2.34) | 0.900 | 0.35 (0.13–1.06) | 0.050 | 0.37 (0.14–1.05) | 0.051 |
| Other benzimidazole derivatives | 8 | 1,708 | Ref. | | Ref. | | Ref. | |
| **Toxic epidermal necrolysis** | | | | | | | | |

(*Continued*)

**Table 3.** (Continued)

| Serious adverse events of interest | Cases* | Non-cases | Analysis 1[a] | | Analysis 2[b] | | Analysis 3[c] | |
|---|---|---|---|---|---|---|---|---|
| | | | ROR (IC95%) | P-value | aROR (IC95%) | P-value | aROR (IC95%) | P-value |
| Benzimidazole derivatives | 8 | 5,129 | 0.35 (0.15–0.69) | 0.005 | 0.32 (0.13–0.69) | 0.007 | 0.26 (0.11–0.55) | <0.001 |
| Other anthelminthic drugs | 62 | 13,869 | Ref. | | Ref. | | Ref. | |
| Albendazole | 5 | 3,416 | 0.84 (0.20–4.08) | 0.800 | 0.45 (0.07–2.88) | 0.400 | 0.54 (0.09–3.38) | 0.500 |
| Other benzimidazole derivatives | 3 | 1,713 | Ref. | | Ref. | | Ref. | |
| **Bullous conditions** | | | | | | | | |
| Benzimidazole derivatives | 49 | 5088 | 0.70 (0.51–0.96) | 0.029 | 0.44 (0.30–0.64) | <0.001 | 0.41 (0.29–0.59) | <0.001 |
| Other anthelminthic drugs | 188 | 13,743 | Ref. | | Ref. | | Ref. | |
| Albendazole | 25 | 3,396 | 0.52 (0.29–0.92) | 0.022 | 0.24 (0.12–0.48) | <0.001 | 0.27 (0.14–0.54) | <0.001 |
| Other benzimidazole derivatives | 24 | 1,692 | Ref. | | Ref. | | Ref. | |
| **Allergic disorders** | | | | | | | | |
| Benzimidazole derivatives | 71 | 5,066 | 1.52 (1.13–2.03) | 0.005 | 1.31 (0.89–1.92) | 0.200 | 1.12 (0.81–1.54) | 0.500 |
| Other anthelminthic drugs | 127 | 13,804 | Ref. | | Ref. | | Ref. | |
| Albendazole | 35 | 3,386 | 0.48 (0.30–0.77) | 0.002 | 0.24 (0.12–0.47) | <0.001 | 0.35 (0.20–0.62) | <0.001 |
| Other benzimidazole derivatives | 36 | 1,680 | Ref. | | Ref. | | Ref. | |
| **Seizures** | | | | | | | | |
| Benzimidazole derivatives | 73 | 5,064 | 2.53 (1.83–3.48) | <0.001 | 1.92 (1.29–2.85) | 0.001 | 1.75 (1.23–2.48) | 0.002 |
| Other anthelminthic drugs | 79 | 13,852 | Ref. | | Ref. | | Ref. | |
| Albendazole | 38 | 3,383 | 0.54 (0.34–0.86) | 0.009 | 0.52 (0.28–0.99) | 0.043 | 0.65 (0.36–1.16) | 0.140 |
| Other benzimidazole derivatives | 35 | 1,681 | Ref. | | Ref. | | Ref. | |
| **Leucopenia** | | | | | | | | |
| Benzimidazole derivatives | 81 | 5,056 | 3.97 (2.83–5.61) | <0.001 | 3.89 (2.64–5.76) | <0.001 | 4.62 (3.24–6.64) | <0.001 |
| Other anthelminthic drugs | 56 | 13,875 | Ref. | | Ref. | | Ref. | |
| Albendazole | 76 | 3,345 | 7.77 (3.48–22.2) | <0.001 | 14.3 (4.25–89.0) | <0.001 | 5.57 (2.37–16.4) | <0.001 |
| Other benzimidazole derivatives | 5 | 1,711 | Ref. | | Ref. | | Ref. | |
| **Marrow depression and hypoplastic anemias** | | | | | | | | |
| Benzimidazole derivatives | 50 | 5,087 | 8.55 (4.98–15.50) | <0.001 | 9.44 (5.01–18.9) | <0.001 | 10.7 (6.01–20.0) | <0.001 |
| Other anthelminthic drugs | 16 | 13,915 | Ref. | | Ref. | | Ref. | |
| Albendazole | 49 | 3,372 | 24.9 (5.46–441) | 0.001 | 11.1 (2.31–199) | 0.019 | 11.9 (2.52–212) | 0.015 |
| Other benzimidazole derivatives | 1 | 1,715 | Ref. | | Ref. | | Ref. | |
| **Headaches** | | | | | | | | |
| Benzimidazole derivatives | 154 | 4,983 | 1.08 (0.89–1.30) | 0.400 | 1.35 (1.08–1.68) | 0.007 | 1.25 (1.01–1.53) | 0.034 |
| Other anthelminthic drugs | 387 | 13,544 | Ref. | | Ref. | | Ref. | |
| Albendazole | 130 | 3,291 | 2.78 (1.83–4.42) | <0.001 | 1.72 (1.00–3.06) | 0.056 | 1.77 (1.08–3.01) | 0.027 |
| Other benzimidazole derivatives | 24 | 1,692 | Ref. | | Ref. | | Ref. | |
| **Dizziness** | | | | | | | | |
| Benzimidazole derivatives | 123 | 5,014 | 1.50 (1.20–1.87) | <0.001 | 1.56 (1.20–2.02) | <0.001 | 1.46 (1.14–1.86) | 0.002 |
| Other anthelminthic drugs | 224 | 13,707 | Ref. | | Ref. | | Ref. | |
| Albendazole | 95 | 3,326 | 1.72 (1.14–2.68) | 0.012 | 0.84 (0.49–1.46) | 0.500 | 0.98 (0.60–1.64) | >0.900 |
| Other benzimidazole derivatives | 28 | 1,688 | Ref. | | Ref. | | Ref. | |

a, crude analysis

b, adjusted analysis on age, sex, reporter type, reporting period and geographical zone

c, adjusted analysis on age, sex, reporter type and geographical zone with imputed data for missing values (KNN method).

* Cases are ICSRs where at least one suspected serious adverse event of interest is reported after the drug exposure. An ICSR can be used in multiple analysis if it contains multiple events of interest.

by decreasing adjusted ROR (aROR): bone marrow failure and hypoplastic anemia (aROR 9.44, 95% CI 5.01–18.9), leukopenia (aROR 3.89, 95% CI 2.64–5.76), hepatic disorders (aROR 3.10, 95% CI 2.59–3.71, hepatitis (aROR 2.88, 95% CI 2.29–3.63), urticaria (aROR 2.02, 95% CI 1.36–2.99), seizures (aROR 1.92, 95% CI 1.29–2.85), dizziness (aROR 1.56, 95% CI 1.20–2.02), nausea and vomiting (aROR 1.50, 95% CI = 1.22–1.83), and headaches (aROR 1.35, 95% CI 1.03–1.68).

As for the secondary analysis comparing ALB to other benzimidazole derivatives, disproportionality analysis revealed an over-reporting of certain serious adverse events, ranked by decreasing adjusted ROR: leukopenia (aROR 14.3, 95% CI 4.25–89.0), bone marrow failure and hypoplastic anemias (aROR 11.1, 95% CI 2.31–199), hepatic disorders (aROR 5.70, 95% CI 3.88–8.59) and hepatitis (aROR 4.82, 95% CI 2.98–8.16). There was no disproportionate reporting for the other adverse events analyzed.

The sensitivity analysis with data imputation showed similar disproportionate reporting than those of the main analyses, except for a new signal of serious headache observed after ALB exposure (aROR 1.77, 95% CI 1.08–3.01).

## Discussion

This research established a detailed assessment of serious adverse events reported after the use of benzimidazole derivatives, using a case/non-case approach, analyzing data collected from the WHO Adverse Drug Reaction database, spanning 2000 to 2023. Our analyses revealed statistically significant pharmacovigilance signals for several serious adverse events, including bone marrow failure, hypoplastic anemia, leukopenia, hepatic disorders and hepatitis, with particularly strong signals reported after ALB exposure. The signal observed for seizures after benzimidazole derivatives exposure was not previously known nor mentioned in official information and constitutes a new pharmacovigilance signal.

While Bagheri *et al.* (2004) described various adverse reactions to anthelmintic drugs recorded in the French pharmacovigilance database from 1988 to 1999, our study is the first to examine serious adverse events associated with benzimidazole derivatives in the WHO pharmacovigilance database through disproportionality analyses [20].

Our main hypothesis, that there were new signals of serious adverse events reported with benzimidazole derivatives compared to other anthelmintic drugs, is supported by our analyses. Our results revealed pharmacovigilance signals, indicated by statistically significant disproportionate reporting, for bone marrow failure and hypoplastic anemia, leukopenia, liver disorders, hepatitis, hepatic disorders, urticaria, convulsions/seizures, dizziness, nausea and vomiting, and headaches in patients exposed to benzimidazole derivatives, with strong signals observed for the first six. Specifically, the analysis within class comparing ALB with other benzimidazole derivatives (i) showed that pharmacovigilance signals for bone marrow failure and hypoplastic anemias, leukopenia, liver disorders, hepatitis, and hepatic disorders were strongly reported after ALB exposure.

Placing these results in the context of existing literature, our findings align with previous studies highlighting the hematological and hepatotoxic risks associated with ALB, corroborating existing safety concerns [4]. Although these effects are rare, Fernandez *et al.* (1996) were the first to report a case of pancytopenia (bone marrow failure) associated with ALB in a 75-year-old woman treated for hepatic hydatidosis [21]. Additionally, Bagheri *et al.* (2004) reported pancytopenia with ALB and MEB treatments as part of a long course of treatment for echinococcosis, with high doses of medicinal products [20]. Xing *et al.* (2018) also reported a case of fatal pancytopenia after 15 days of ALB treatment [22]. Several studies are consistent with our findings of disproportionality observed for the hepatotoxic risks associated with ALB

[23–29]. Although ALB excellent safety profile underpins mass drug distribution campaigns, it is essential to remain aware of and vigilant about the serious adverse reactions it can potentially induce. As the signals highlighted in our study were based on findings from existing data, it would be valuable to conduct a real-world pharmacoepidemiologic study to confirm these signals.

The pathophysiological mechanisms underlying the suspected adverse events following exposure to benzimidazole derivatives remain incompletely understood. For hematological disorders such as bone marrow failure, hypoplastic anemia, leukopenia, and pancytopenia, the primary hypothesis is that this class of drugs can inhibit tubulin polymerization, potentially impairing cell turnover and leading to a depletion of cell lineages [21]. Additionally, benzimidazoles may interfere with folate metabolism, further contributing to hematological toxicity [30]. Hepatic disorders, which are well-known adverse drug reactions, might be linked to the drug's metabolites, primarily eliminated via the biliary route [4]. The mechanisms underlying seizures and skin disorders are unclear and need further investigations. One hypothesis is that these effects might involve the direct action of the drug on parasites in the blood (such as loiasis) or in brain (such as taeniasis), provoking allergic and/or neurologic reactions due to parasite death, circulation of their remains, or embolization in brain capillaries. The limitations of this study include the under-reporting of suspected adverse reactions (and the potential differences in this under-reporting between countries), as well as the lack of information on drug exposure in the population. These limitations, inherent in studies using pharmacovigilance databases [31,32], restrict the measurement of the true incidence of adverse reactions. Although under-reporting may be less significant since we focused on serious adverse events (which are more likely to be reported) [33], our analyses cannot measure the actual risk of adverse reactions but only differences in reported events. Indeed, the subjects in our "control" (non-case) group are not healthy controls, but patients who have reported other adverse events. Given the multiplicity of statistical analyses performed, which increases the risk of Type I errors, we did not apply formal corrective methods for multiple comparisons. However, we focused on 17 specific adverse events selected based on a comprehensive literature review and preliminary frequency analysis in our dataset. The consistent findings across various sub-analyses suggest that the risk of random statistically significant results is low. Moreover, it is crucial to consider the potential for protopathic bias in interpreting our results. Several adverse effects, such as liver disorders, seizures, or urticaria can also be symptoms of parasitic infections for which benzimidazole derivatives are prescribed. This could lead to misattribution of disease symptoms to the drug, potentially inflating the apparent association with benzimidazole use.). Another limitation is the possibility of indication bias: not all anthelmintic drugs are used in the same situations, and the signals found with benzimidazole derivatives could be partly explained by these differences. Unfortunately, the precise indication was not available in the database and could not be considered. However, indication bias is less important when comparing drugs in the same therapeutic area (which we did) than when comparing the drugs of interest to all other drugs reported in the database across all therapeutic areas and indications. Additionally, by eliminating reports recording the concomitant role of our drugs, we have reduced the impact of competition bias on the drug-event combination of interest [34]. Nevertheless, these results must be interpreted with caution due to potentially missing information.

The main strength of this study is the use of the WHO VigiBase database, which enables the study of adverse events reported worldwide and provides important information on the use of drugs in real-life situations. Our study is the first to evaluate the serious adverse events of benzimidazole derivatives in the WHO database using a disproportionality analysis. Moreover, the detection of emerging signals of serious adverse events, such as bone marrow failure,

hypoplastic anemias, leukopenia, and hepatic disorders, is paramount to ensure patient safety. By comparing ALB with other benzimidazole derivatives, and more specifically with MEB, the study provides a detailed assessment of the specific risks associated with each drug within the same therapeutic class. This makes it possible to identify risks specific to certain drugs, rather than the entire class.

The discovery of statistically significant disparities for several serious adverse events, with particularly strong associations attributed to ALB, confirms the reliability of the results and highlights safety issues specific to this drug. The revelation of a significant association between ALB and headache, which had not been detected in the main analysis, underscores the value of the analysis conducted, capable of detecting potentially infrequent but clinically relevant adverse events. Finally, this research builds upon previous studies, such as those by Bagheri *et al.* (2004), by providing recent and extensive data on the adverse reactions of anthelmintic drugs, focusing specifically on benzimidazole derivatives [20].

## Conclusion

This first study to disproportionate reporting of adverse events reported with benzimidazole derivatives in the WHO database, complements and extends previous work carried out on smaller national databases, suggests that ALB presents an increased risk for several serious adverse events compared to other benzimidazole derivatives. It also revealed new potential pharmacovigilance signal for benzimidazole derivatives: seizures. These results underline the need for constant vigilance and monitoring for patients taking benzimidazole derivatives, particularly ALB. These results call for further studies to confirm these links and examine the mechanisms underlying these adverse effects.

## Supporting information

**S1 Material. Review of the case reports in the literature.**
(DOCX)

**S1 Table. Adverse events of interest and corresponding MedDRA terms.**
(DOCX)

**S2 Table. Reported MedDRA System Organ Class (SOC) according to anthelmintic drugs type.**
(DOCX)

## Author Contributions

**Conceptualization:** Jean-Luc Faillie, Jérémy T. Campillo.

**Data curation:** Pamella Modingam, Jérémy T. Campillo.

**Formal analysis:** Pamella Modingam.

**Investigation:** Pamella Modingam, Jérémy T. Campillo.

**Methodology:** Pamella Modingam, Jean-Luc Faillie.

**Project administration:** Jean-Luc Faillie, Jérémy T. Campillo.

**Resources:** Jean-Luc Faillie, Jérémy T. Campillo.

**Supervision:** Jean-Luc Faillie, Jérémy T. Campillo.

**Validation:** Jean-Luc Faillie, Jérémy T. Campillo.

**Visualization:** Pamella Modingam.

**Writing – original draft:** Pamella Modingam.

**Writing – review & editing:** Pamella Modingam, Jean-Luc Faillie, Jérémy T. Campillo.

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
