## [Decision Letter · Decision Letter 0]

9 Sep 2024

Dear Dr Campillo,

Thank you very much for submitting your manuscript "Serious adverse drug reactions associated with benzimidazole derivatives: a disproportionality analysis from the World Health Organization's pharmacovigilance database" for consideration at PLOS Neglected Tropical Diseases. As with all papers reviewed by the journal, your manuscript was reviewed by members of the editorial board and by several independent reviewers. In light of the reviews (below this email), we would like to invite the resubmission of a significantly-revised version that takes into account the reviewers' comments. 

The reviewers have acknowledged the novelty of your study, particularly the use of the World Health Organization's pharmacovigilance database to analyze benzimidazole derivatives. They noted that the manuscript is well-written and includes valuable sub-analyses. The objectives are clearly articulated, and the hypothesis is testable.

However, there are areas that require further attention:

Concerns were raised about the adequacy of the sample size in relation to the power of the analysis.

While the figures and tables are generally clear, there may be room for improvement in their presentation.

Given these points, the manuscript is being returned with a recommendation for Major Revision. Please address the reviewers' comments in detail and resubmit your revised manuscript for further consideration.

We look forward to receiving your revised submission.

We cannot make any decision about publication until we have seen the revised manuscript and your response to the reviewers' comments. Your revised manuscript is also likely to be sent to reviewers for further evaluation.

Sincerely,

Eduardo José Lopes-Torres, Ph.D.

Academic Editor

Jong-Yil Chai

Section Editor

Dear Authors,

Thank you for submitting your manuscript titled "Serious adverse drug reactions associated with benzimidazole derivatives: a disproportionality analysis from the World Health Organization's pharmacovigilance database" to PLOS Neglected Tropical Diseases. After careful consideration and review by our expert reviewers, we have reached a decision.

The reviewers have acknowledged the novelty of your study, particularly the use of the World Health Organization's pharmacovigilance database to analyze benzimidazole derivatives. They noted that the manuscript is well-written and includes valuable sub-analyses. The objectives are clearly articulated, and the hypothesis is testable.

However, there are areas that require further attention:

Concerns were raised about the adequacy of the sample size in relation to the power of the analysis.

While the figures and tables are generally clear, there may be room for improvement in their presentation.

Given these points, the manuscript is being returned with a recommendation for Major Revision. Please address the reviewers' comments in detail and resubmit your revised manuscript for further consideration.

We look forward to receiving your revised submission.

Best regards,

Eduardo Torres

Academic Editor PLOS Neglected Tropical Diseases

Reviewer's Responses to Questions

**Key Review Criteria Required for Acceptance?**

**Methods**

-Are the objectives of the study clearly articulated with a clear testable hypothesis stated?

-Is the study design appropriate to address the stated objectives?

-Is the population clearly described and appropriate for the hypothesis being tested?

-Is the sample size sufficient to ensure adequate power to address the hypothesis being tested?

-Were correct statistical analysis used to support conclusions?

-Are there concerns about ethical or regulatory requirements being met?

Reviewer #1: (No Response)

Reviewer #2: Yes

**Results**

-Does the analysis presented match the analysis plan?

-Are the results clearly and completely presented?

-Are the figures (Tables, Images) of sufficient quality for clarity?

Reviewer #1: (No Response)

Reviewer #2: Yes

**Conclusions**

-Are the conclusions supported by the data presented?

-Are the limitations of analysis clearly described?

-Do the authors discuss how these data can be helpful to advance our understanding of the topic under study?

-Is public health relevance addressed?

Reviewer #1: (No Response)

Reviewer #2: Yes

**Editorial and Data Presentation Modifications?**

Reviewer #1: (No Response)

Reviewer #2: (No Response)

**Summary and General Comments**

Reviewer #1: (No Response)

Reviewer #2: The authors reported the results of a disproportionality analysis using the World Health Organization's pharmacovigilance database regarding benzimidazole derivatives for the first time. The manuscript is well written with several subanalyses.

I only had few comments:

- In the Discussion section, the authors stated “severe leukopenia, severe liver disorders, hepatitis, severe hepatic disorders, severe urticaria, severe convulsions/seizures, dizziness, severe nausea and vomiting”, which is misleading as the ADR in the Vigibase are “serious” and not “severe”. Indeed, an ADR could be considered serious in the Vigibase if it is a “medically important event or reaction”, independent of any notion of severity.

- The authors could elaborate on possible pathophysiological mechanisms at the origin of the ADR found, even if this not the purpose of a disproportionality analysis.

- The authors should discuss possible protopathic bias, as several ADR found can be symptoms of parasitic infections (liver disorders, seizures, urticaria…)

- The authors should provide further details on the literature review: which “expert pharmacovigilants”? which keywords? which search engines?…

- Some corrections need to be done: typos (a new signals, as well-tolerated, Table 2 shows describes, according anthelminthic drugs,…) ; inconsistencies (in abstract: aROR sometimes written, sometimes not) ; “other benzimidazole derivatives (i)” in the Discussion section?; legend 1 not explained in Supplementary Table 2; article 22 misquoted ; “of which 5,131 cases” instead of 5,137.

PLOS authors have the option to publish the peer review history of their article (what does this mean?). If published, this will include your full peer review and any attached files.

Reviewer #1: No

Reviewer #2: No
---

## [Decision Letter · Decision Letter 1]

16 Oct 2024

Dear Dr Campillo,

We are pleased to inform you that your manuscript 'Serious adverse events reported with benzimidazole derivatives: a disproportionality analysis from the World Health Organization's pharmacovigilance database' has been provisionally accepted for publication in PLOS Neglected Tropical Diseases.

Before your manuscript can be formally accepted, you will need to **address the changes suggested by Reviewer 1** and and complete some formatting changes, which you will receive in a follow up email. A member of our team will be in touch with a set of requests.

Best regards,

Eduardo José Lopes-Torres, Ph.D.

Academic Editor

Jong-Yil Chai

Section Editor

Reviewer's Responses to Questions

**Key Review Criteria Required for Acceptance?**

**Methods**

-Are the objectives of the study clearly articulated with a clear testable hypothesis stated?

-Is the study design appropriate to address the stated objectives?

-Is the population clearly described and appropriate for the hypothesis being tested?

-Is the sample size sufficient to ensure adequate power to address the hypothesis being tested?

-Were correct statistical analysis used to support conclusions?

-Are there concerns about ethical or regulatory requirements being met?

Reviewer #1: There is an improvement in the method section. However, here are some additional suggestions to improve the manuscript

Objective (introduction and abstract)

L28 & L99: It is more appropriate to say “potential pharmacovigilance signal”. Disproportionality analysis allows to identify potential quantitative signals. Further assessment is required to confirm as signals.

Methods

L105 states “Deduplicated data were extracted from … VigiBase”. You responded to reviewer that you “did not use the VigiLyze tool for data extraction. Instead, a de-duplicated dataset was specifically provided by the Uppsala Monitoring Centre upon request.” This last information needs to appear in the method section. It can explain discrepancies in case some readers extract data from VigiLyze for comparison.

L132 – 136: I would strongly suggest to remove these lines from Case definition and put them in the data source section. This will allow to focus on L136 to 146 for case definition.

Reviewer #2: Yes

**Results**

-Does the analysis presented match the analysis plan?

-Are the results clearly and completely presented?

-Are the figures (Tables, Images) of sufficient quality for clarity?

Reviewer #1: The results are clearly presented.

Reviewer #2: Yes

**Conclusions**

-Are the conclusions supported by the data presented?

-Are the limitations of analysis clearly described?

-Do the authors discuss how these data can be helpful to advance our understanding of the topic under study?

-Is public health relevance addressed?

Reviewer #1: The conclusions are supported by the data presented. The limitations have been discussed as well as the relevance of these findings.

However, the conclusion should be nuanced to present seizure as a new potential pharmacovigilance signal, since signal assessment was not part of this manuscript.

Reviewer #2: Yes

**Editorial and Data Presentation Modifications?**

Reviewer #1: (No Response)

Reviewer #2: (No Response)

**Summary and General Comments**

Reviewer #1: (No Response)

Reviewer #2: All comments have been made.

Please note the typo in supplementary materials: Encephlopathy

PLOS authors have the option to publish the peer review history of their article (what does this mean?). If published, this will include your full peer review and any attached files.

Reviewer #1: No

Reviewer #2: No

---

## [Editor Report · Acceptance letter]

30 Oct 2024

Dear Dr Campillo,

We are delighted to inform you that your manuscript, "Serious adverse events reported with benzimidazole derivatives: a disproportionality analysis from the World Health Organization's pharmacovigilance database," has been formally accepted for publication in PLOS Neglected Tropical Diseases.

Best regards,

Shaden Kamhawi

co-Editor-in-Chief

Paul Brindley

co-Editor-in-Chief
